# Persistent $CO_2$ emissions and hydrothermal unrest following the 2015 earthquake in Nepal

Frédéric Girault [1], Lok Bijaya Adhikari[2], Christian France-Lanord [3], Pierre Agrinier[4], Bharat P. Koirala[2], Mukunda Bhattarai[2], Sudhan S. Mahat[5], Chiara Groppo [6], Franco Rolfo [6], Laurent Bollinger[7] & Frédéric Perrier[1]

Fluid–earthquake interplay, as evidenced by aftershock distributions or earthquake-induced effects on near-surface aquifers, has suggested that earthquakes dynamically affect permeability of the Earth's crust. The connection between the mid-crust and the surface was further supported by instances of carbon dioxide ($CO_2$) emissions associated with seismic activity, so far only observed in magmatic context. Here we report spectacular non-volcanic $CO_2$ emissions and hydrothermal disturbances at the front of the Nepal Himalayas following the deadly 25 April 2015 Gorkha earthquake (moment magnitude $M_w = 7.8$). The data show unambiguously the appearance, after the earthquake, sometimes with a delay of several months, of $CO_2$ emissions at several sites separated by > 10 kilometres, associated with persistent changes in hydrothermal discharges, including a complete cessation. These observations reveal that Himalayan hydrothermal systems are sensitive to co- and post-seismic deformation, leading to non-stationary release of metamorphic $CO_2$ from active orogens. Possible pre-seismic effects need further confirmation.

[1] Physics of Natural Sites, Institut de Physique du Globe de Paris, Sorbonne Paris Cité, Université Paris Diderot, CNRS UMR 7154, 1 rue Jussieu, F-75005 Paris, France. [2] Department of Mines and Geology, National Seismological Centre, Lainchaur, Kathmandu, Nepal. [3] Centre de Recherches Pétrographiques et Géochimiques, Université de Nancy, CNRS UMR 7358, F-54500 Vandoeuvre-lès-Nancy, France. [4] Stable Isotopes Geochemistry, Institut de Physique du Globe de Paris, Sorbonne Paris Cité, Université Paris Diderot, CNRS UMR 7154, 1 rue Jussieu, F-75005 Paris, France. [5] Sanjen Jalavidhyut Company Limited, Lazimpat, Kathmandu, Nepal. [6] Department of Earth Sciences, IGG-CNR, University of Turin, Via Verdi, 8, I-10124 Turin, Italy. [7] CEA, DAM, DIF, F-91297 Arpajon, France. Correspondence and requests for materials should be addressed to F.G.(email: girault@ipgp.fr)

Understanding fluid–earthquake interplay has long received a lot of attention[1]. For the last 20 years, there has been a growing number of evidence for fluid-driven earthquakes[2]. Deep fluids have been shown controlling aftershock distribution in the tectonic contexts of rifting[3–5], subduction[6,7], and reverse[8] and strike-slip faulting[9]. Contemporaneously, numerous observations of earthquake-induced effects on near-surface aquifers have been reported[10], including mainly changes in stream and spring discharge[11–13], groundwater level[14,15] and temperature[16–19] in various tectonic contexts. These observations have suggested that earthquakes dynamically affect permeability of the Earth's crust[20,21]. Carbon dioxide ($CO_2$) emissions were observed in association with seismicity in the case of the Matsushiro swarm[9] in Japan or, recently, at the Lassen volcano[22] (Cascades Range, USA) and in the Eger Rift[23] (Czech Republic), which suggests connection between the mid-crust and the ground surface through gas transport. However, such examples so far were only observed in the presence of magmatic activity.

The Himalayan orogen results from the India–Eurasia collision[24], the Main Himalayan Thrust (MHT) accommodating at 2 cm year$^{-1}$ half of the shortening between the two continents[25]. The largest earthquake in Nepal before 2015, the 1934 Bihar-Nepal earthquake (moment magnitude $M_w \sim 8.2$) in Eastern Nepal, which claimed >15,000 lives, ruptured the MHT up to the surface over 150 kilometres along the Main Frontal Thrust[26]. Inter-seismic deformation is associated with intense background seismicity[27], with 4–5 events of local magnitude $M_L > 5$ per year,

concentrated between 10 and 25 kilometres depth at the foot of the Himalayan topographic rise[28]. This region – the Main Central Thrust (MCT) zone[29] – also exhibits numerous hydrothermal systems[30] (Fig. 1). Following evidence of degassing from chemical and isotopic analysis of hot springs and rivers[31,32], large $CO_2$ emissions were discovered near hot springs[30,33], with $CO_2$ fluxes at places similar to diffusive fluxes from active volcanoes[30,34]. The seasonal and yearly stability of the soil–gas radon concentration time-series[34,35], the invariant radon–$CO_2$ fluxes relationship[30,36], and the results of watering experiments[36] at selected sites attest to the remarkable temporal stability of these hydrothermal systems, even during monsoon. These non-volcanic $CO_2$ emissions are characterised by radiogenic helium, high radon content, and carbon isotopic compositions suggesting metamorphic $CO_2$ production at >5 kilometres depth[30–32,34]. In the accepted conceptual model[31,32], from the decarbonation source at >5 kilometres depth, $CO_2$ percolates through fracture networks in the MCT zone, where it mixes with meteoric water. Near the water table, a fraction of $CO_2$ may degas, and water discharges eventually at the surface as a hot spring. Degassed $CO_2$ may also be transported faster to the surface through a network of interrelated faults without interaction with hydrothermal circulations[37].

The $M_w = 7.8$ 25 April 2015 Gorkha earthquake[38] (Fig. 1) caused >9000 deaths, with fatality rates >1% in mountainous areas north of Kathmandu[39], reaching 100% at some places[40]. It partly ruptured the MHT along a 120-kilometre-long segment east from the epicentre[41], whose northern limit coincides with the

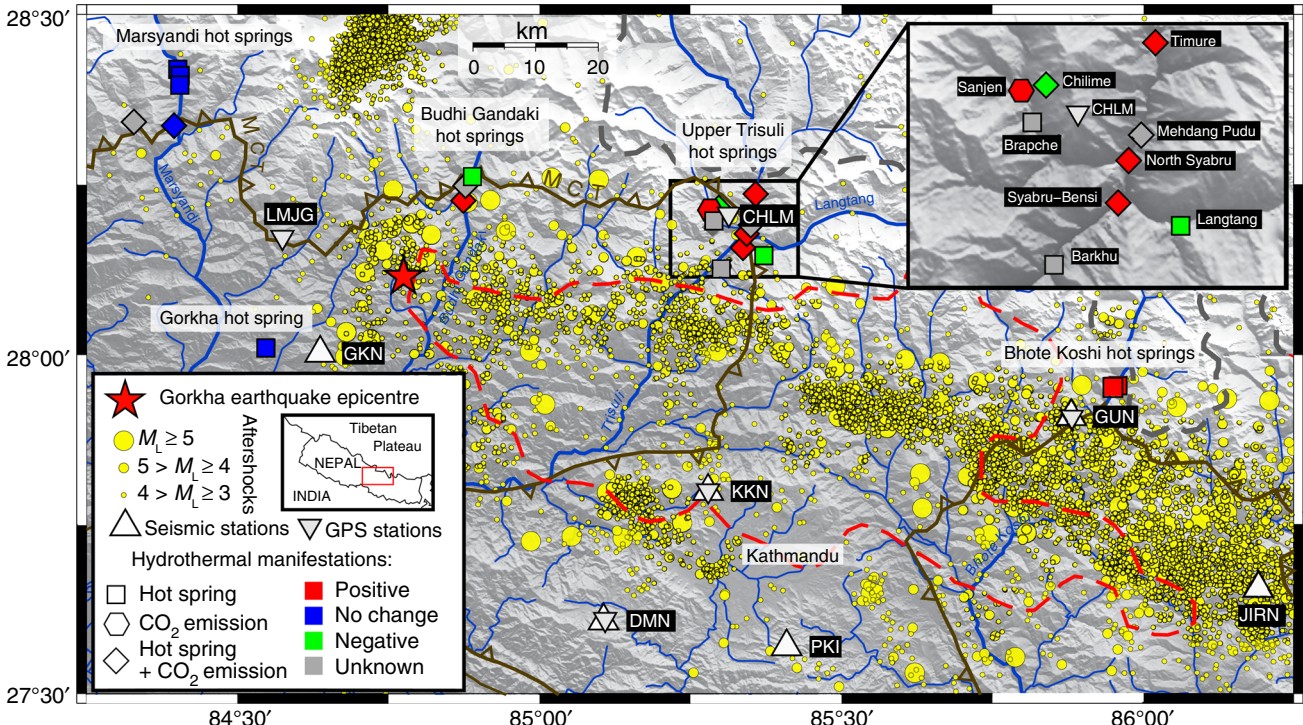

**Fig. 1** Post-seismic responses of carbon dioxide emissions and hot springs following the 25 April 2015 Gorkha earthquake in Central Nepal. Post-seismic responses are shown as coloured symbols: positive effect (red), negative effect (green), absence of any significant change (blue), or when it is unknown (grey). Aftershock epicentres ($M_L \geq 3$) from 25 April 2015 to 31 December 2017 (Nepal National Seismological Centre, NSC) are plotted as yellow dots[43]. The Main Central Thrust (MCT) shear zone (in dark grey), which branches at depth on the MHT, separates low-grade Lesser Himalayan Sequence (LHS) and high-grade Greater Himalayan Sequence (GHS) to the north. The red dashed contour shows the area of slip larger than two metres[41]. The bottom left inset shows the location of the main study area. The top right inset depicts an enlargement of the Upper Trisuli valley. Seismic stations belong to the network operated by NSC (Nepal) and DASE (France): Gorkha (GKN), Daman (DMN), Kakani (KKN), Phulchoki (PKI), Gumba (GUN) and Jiri (JIRN). Global Positioning System (GPS) stations belong to the Nepal Geodetic network operated by Caltech (USA), DMG (Nepal) and DASE (France): Lamjung (LMJG), Daman (DMN), Chilime (CHLM), Kakani (KKN) and Gumba (GUN). Details of the main $CO_2$ emission sites and of the hot spring sites as well as their respective responses to the earthquake are given in Table 1 and Supplementary Table 1 and in Table 2 and Supplementary Table 2, respectively. Figure performed using Generic Mapping Tool (http://gmt.soest.hawaii.edu)

**Table 1 Post-seismic effects on the carbon dioxide emissions at five hydrothermal sites in Central Nepal. Data of $CO_2$ flux, total $CO_2$ discharge, and carbon isotopic ratio of $CO_2$ emissions are compiled before and after the Gorkha earthquake**

| Site | Location | Before/after Gorkha earthquake | $CO_2$ flux (g m$^{-2}$ d$^{-1}$) | | | $CO_2$ discharge ($10^{-3}$ mol s$^{-1}$) | $\delta^{13}C$ of $CO_2$ (V-PDB) (‰) | | Post-seismic effect |
|---|---|---|---|---|---|---|---|---|---|
| | | | $N_{meas}$ ($n_{point}$) | Min/max range | Geometric mean | | $N_{meas}$ ($n_{point}$) | Geometric mean | |
| *Budhi Gandaki Valley* | | | | | | | | | |
| Machhakhola | Near Hot Spring | After | 63(43) | 21.6/81,800 | 514 ± 16 | >1240 ± 330 | 4(4) | −3.2 ± 0.2 | Increase |
| *Upper Trisuli Valley* | | | | | | | | | |
| Sanjen | Tunnel | After | 36(33) | 13.2/94,700 | 640 ± 40 | >580 ± 150 | 4(4) | −0.1 ± 0.2 | Increase |
| | Piezometer DH1 | After | 3(1) | | 92,900 ± 5200 | n.e. | 3(1) | −0.94 ± 0.01 | Increase |
| | Piezometer DH2 | After | 3(1) | | 15,400 ± 2900 | n.e. | 4(1) | −0.70 ± 0.01 | Increase |
| Chilime | Gas Zone | Before[a,c] | 192 (92) | 3.4/123,000 | 608 ± 12 | 470 ± 120 | 2(2) | −1.5 ± 0.1 | Decrease |
| | | After (01/2016) | 138 (131) | 2.6/10,100 | 136 ± 2 | 91 ± 19 | 3(3) | −1.50 ± 0.06 | |
| | | After (01/2018) | 99(93) | 0.3/2620 | 21.3 ± 2.0 | 19 ± 4 | 0 | n.m. | |
| Syabru-Bensi | GZ1-2 Terrace | Before[a,d,e] (2006–2011) | 652 (333) | 2.5/236,000 | 196 ± 2 | 480 ± 50 | 7(3) | −0.88 ± 0.07 | Increase |
| | | After (2015–2018) | 577 (378) | 2.8/226,000 | 236 ± 3 | 1010 ± 110 | 14(4) | −0.74 ± 0.02 | |
| | Hot Spring FF2 | After | 0 | | | n.e. | 2(1) | −2.2 ± 0.1 | Increase |
| Timure | Northern Profile | Before[c] | 69(44) | 73/5600 | 767 ± 22 | 91 ± 23 | 1(1) | −1.6 ± 0.1 | Increase |
| | | After (01/2016) | 75(69) | 110/17,400 | 1850 ± 50 | 270 ± 70 | 2(2) | −2.11 ± 0.02 | |
| | | After (09/2017) | 117(89) | 6.1/175,000 | 2170 ± 50 | 740 ± 200 | 0 | n.m. | |

n.e.: not estimated, n.m.: not measured, $N_{meas}$($n_{point}$): total number of measurements (total number of measurement points)
Data from: [a]ref.[30], [b]ref.[52], [c]ref.[51], [d]ref.[34], [e]ref.[33] and [62]

MCT zone (Fig. 1), releasing only partially the stored elastic energy sufficient to produce a possible $M_w$~ 9 mega-earthquake[42]. The mainshock was followed by an intense aftershock sequence[43], with >90 ($M_L > 5$) events over a year. From 25 April 2015 to 31 December 2017, >10,200 ($M_L > 3$), >1100 ($M_L > 4$) and >135 ($M_L > 5$) earthquakes occurred in Central Nepal (Fig. 1), leading to the most important seismic crisis in Nepal since 1934. The Gorkha earthquake is the first large Himalayan earthquake with seismic[43] and Global Positioning System[44,45] (GPS) data, as well as prior data on hydrothermal systems and $CO_2$ emissions[30].

In this paper, we report spectacular outbursts of $CO_2$ and hydrothermal disturbances at several sites in Central Nepal separated by >10 kilometres that have been triggered by the 2015 Gorkha earthquake. These observations, in particular the first earthquake-induced gaseous changes in the absence of magmatic activity, reveal high sensitivity of Himalayan hydrothermal systems to co-, post- and possibly pre- seismic deformation, and non-stationary release of metamorphic $CO_2$ from active orogens.

## Results

**Outbursts of $CO_2$ triggered by the Gorkha earthquake.** New $CO_2$ emissions were observed after the Gorkha earthquake, sometimes spatially associated with substantial changes from the pre-existing $CO_2$ emissions (Table 1 and Supplementary Table 1). Because a large number of $CO_2$ flux measurements (see Methods) were made before ($n = 1718$; December 2006 – January 2011) and after the earthquake ($n = 1053$; November 2015 – January 2018), a quantitative comparison of $CO_2$ emissions before and after the earthquake can be undertaken in the Upper Trisuli valley. Spectacular effects were observed first in Syabru-Bensi, 56 kilometres east from the epicentre (Fig. 1), a site that had been extensively studied from 2006 to 2011. The main gas emission zone, located above the hot springs (Fig. 2; Supplementary Fig. 1), on the west bank of the Trisuli river, changed substantially. Significant $CO_2$ fluxes and discharges appeared near the dwellings above the previously existing gas zone, in a cultivated area where flux level

was close to local background levels before 2009. Hydrogen sulphide is now frequently smelled inside the houses. Total $CO_2$ discharge in the area now reaches $1010 \pm 110$ mmol s$^{-1}$ ($3.8 \pm 0.4$ ton d$^{-1}$), corresponding to an increase factor of $2.1 \pm 0.3$ (Fig. 2). Fluxes at two reference locations (K+6 and K+12; Fig. 2), regularly monitored since November 2015 (Fig. 3), peaked in November 2016 and, in January 2018, continued to remain 2-order of magnitude higher than their pre-seismic values. By contrast, the cavity characterised by the largest pre-earthquake $CO_2$ fluxes ( $>10^5$ g m$^{-2}$ d$^{-1}$) showed one-order-of-magnitude smaller $CO_2$ emission after the earthquake (Supplementary Fig. 4). The cavity fluxes returned to pre-earthquake values in January 2018, more than 2.7 years after the mainshock.

New $CO_2$ emissions and changes in pre-existing $CO_2$ emissions were also observed at other locations in the same valley (Supplementary Fig. 5). In Timure, nine kilometres to the north, $CO_2$ emissions tripled after the earthquake (in November 2015 and January 2016) along a profile which had been measured precisely at different times before 2011 (Supplementary Fig. 6). Repeated measurements in September 2017 yielded a post-seismic $CO_2$ emission increase of a factor of $8 \pm 3$ compared with pre-earthquake values. By contrast, at the Chilime hot spring site, seven kilometres to the west of Timure (Fig. 1), while $CO_2$ fluxes remain significant, the total discharge is reduced by a factor of $5.2 \pm 1.7$ in January 2016, and by a factor of $25 \pm 8$ in January 2018. The $CO_2$ emissions in Timure and Chilime thus depict opposite and unstabilised post-seismic responses > 2.7 years after the mainshock.

Two kilometres to the west of Chilime, at the Sanjen hydropower construction site (Fig. 1), spectacular bubbling was suddenly noticed, beginning of November 2015, in two 40-metre-deep piezometers whose water level was being monitored. When construction resumed during 2016, $CO_2$ emissions appeared at several locations in a tunnel being excavated nearby (Supplementary Fig. 7 and Movie 1). The $CO_2$ emission in the tunnel was still present in January 2018. The $CO_2$ concentration in the air of the tunnel then ranged from 4 to

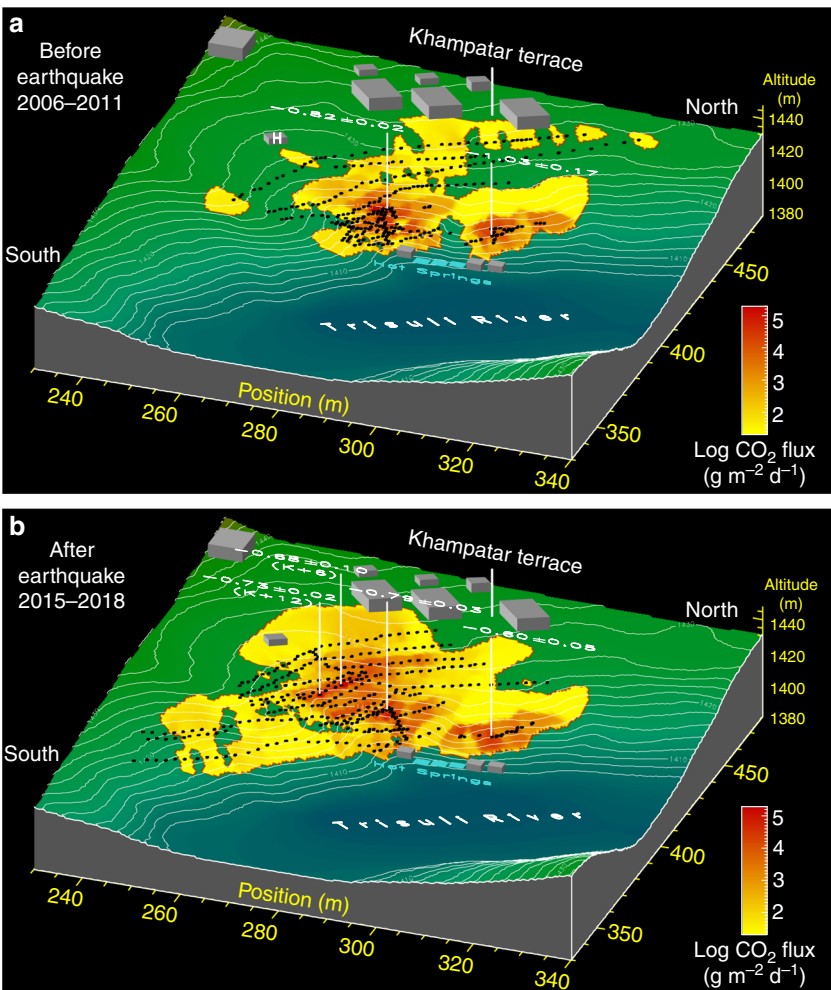

**Fig. 2** Outburst of carbon dioxide emissions following the 2015 Gorkha earthquake in Syabru-Bensi, Upper Trisuli valley. Interpolated maps of surface $CO_2$ fluxes above the main hot springs are shown: **a** before the Gorkha earthquake (652 measurements[33,34] from 2006 to 2011); **b** after the earthquake (577 measurements from November 2015 to January 2018). $CO_2$ fluxes are expressed in logarithmic scale. Topography was determined in 2007 by geodetic levelling[62]. Numbers are carbon isotopic ratios of the gaseous $CO_2$ ($\delta^{13}C$, relative to V-PDB). Total $CO_2$ discharge amounts to $480 \pm 50$ mmol s$^{-1}$ ($1.8 \pm 0.2$ ton d$^{-1}$) before the earthquake and $1010 \pm 110$ mmol s$^{-1}$ ($3.8 \pm 0.4$ ton d$^{-1}$) after, giving a post-seismic overall increase of a factor of $2.1 \pm 0.3$. The surface area of high $CO_2$ emission ( $> 500$ g m$^{-2}$ d$^{-1}$) increased by a factor of about 1.9, with the barycentre shifting eastward by $15 \pm 3$ metres. Example of $CO_2$ fluxes measured along the K profile in the new high emission zone is shown in Supplementary Fig. 2. Radon-222 fluxes were also measured on the terrace and were compared with data obtained before the earthquake. No significant change was found in the $CO_2$–radon fluxes relationship, suggesting similar gas transport mechanism, source, and travel time before and after the earthquake (Supplementary Fig. 3). While no measurement was performed before and during the earthquake, locals reported the unusual simultaneous death of hen and chicken possibly weeks before the earthquake near house H. Figure performed using PV-WAVE® software (Rogue Wave)

5 vol%, thus creating a major health hazard, and the total $CO_2$ discharge was estimated in the tunnel to $580 \pm 150$ mmol s$^{-1}$ ($2.2 \pm 0.6$ ton d$^{-1}$), hence of the same order as the main Syabru-Bensi discharge.

In the Budhi Gandaki valley, 16.5 kilometres to the epicentre (Fig. 1), the highest $CO_2$ emissions ever reported in Central Nepal were observed in January 2017 on the partly flooded riverbank, where no phenomenon had been known to the locals before the earthquake (Supplementary Movie 2). The $CO_2$ emission was still present in January 2018, with a total discharge higher than $1240 \pm 330$ mmol s$^{-1}$ ($4.7 \pm 1.3$ ton d$^{-1}$), similar to the whole $CO_2$ discharge observed in Syabru-Bensi. In addition to the flux from the bank, an innovative method was used to measure the $CO_2$ flux from and through the hot water pond (see Methods and Supplementary Fig. 8). This is the most spectacular new $CO_2$ emission observed so far in the Himalayas and elsewhere in the absence of volcanic activity. Together with the Upper Trisuli

valley (Syabru-Bensi, Sanjen and Timure), such a persistent post-seismic $CO_2$ outburst is unusual. For instance, small outbursts following the 2008 Wenchuan earthquake in China lasted only a few months[46]. In the Marsyandi valley, further west (Fig. 1), by contrast, no comparable phenomenon was observed and peak $CO_2$ fluxes (5300 to 28,700 g m$^{-2}$ d$^{-1}$), while significant, are smaller than in the Upper Trisuli valley.

All these $CO_2$ emissions were quantified in the field during the dry season or in absence of rain, as shown by the 2015–2017 rainfall data in Dhunche (seven kilometres southwest to Syabru-Bensi) and Timure (Fig. 3). At a given site, the $CO_2$ flux data were obtained before and after the earthquake at about the same periods under similar meteorological conditions. Besides, the observed changes in the $CO_2$ emission were done contemporary at several sites separated by more than 10 kilometres. All these care and observations preclude to these changes any environmental, meteorological or shallow origin.

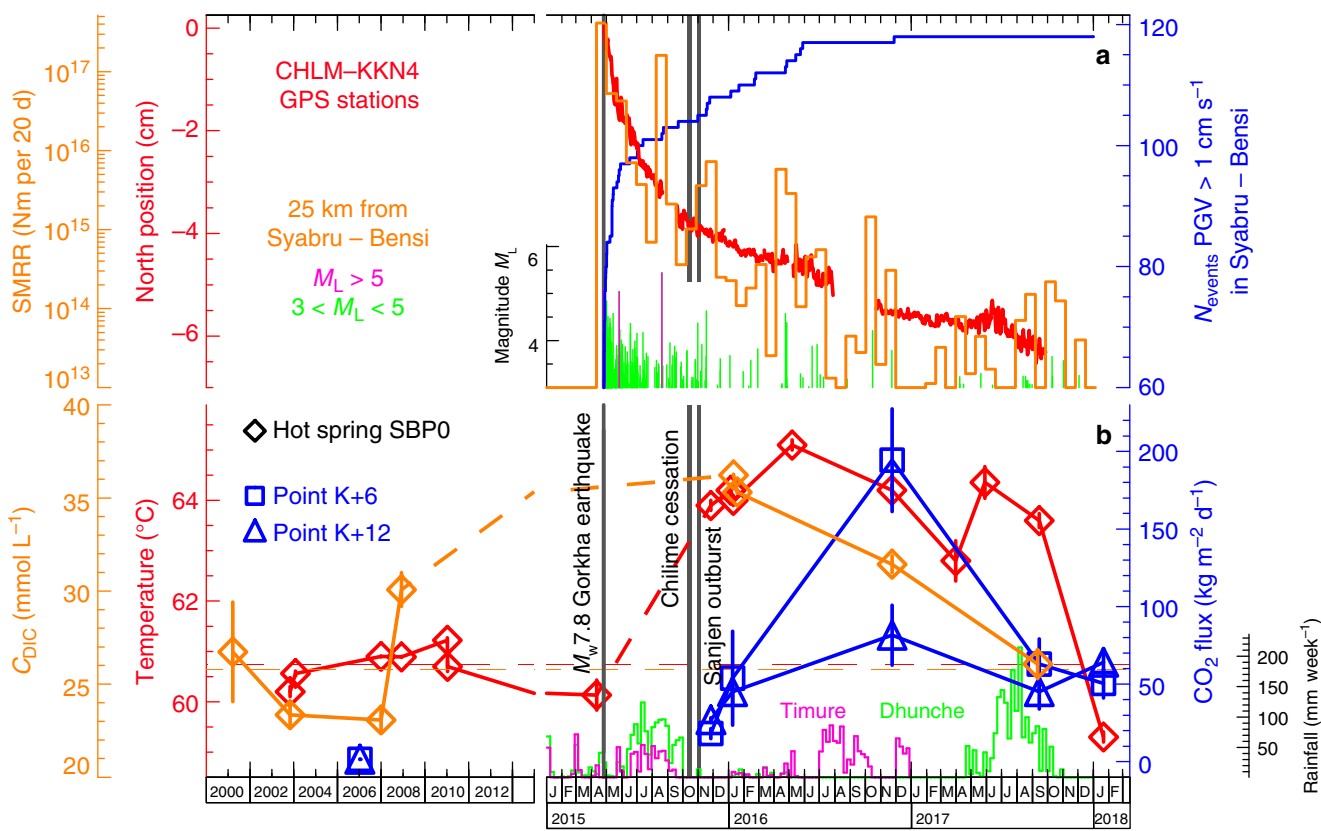

**Fig. 3** Temporal changes in selected parameters in Syabru-Bensi, Upper Trisuli valley. **a** Corrected northward position difference of GPS stations CHLM minus KKN4 (red), cumulated number of aftershocks with vertical Peak Ground Velocity (PGV) higher than 1 cm s$^{-1}$ in Syabru-Bensi (blue) (Supplementary Fig. 12), Seismic Moment Release Rate (SMRR) (orange) and local magnitude (green and purple) of earthquakes ≤ 25 kilometres from Syabru-Bensi. The CHLM minus KKN4 difference of the vertical position helps cancelling most of the seasonal variations. Aftershock epicentres from 25 April 2015 to 31 December 2017 (NSC) are taken into account. **b** Water temperature (red) and dissolved inorganic carbon (DIC) concentration (orange) of the main hot spring of Syabru-Bensi (SBP0) from 2001 to 2018, surface $CO_2$ fluxes (blue) at two locations on the alluvial terrace above the hot spring, K+6 and K+12 (Fig. 2 and Supplementary Fig. 2), and the 2015–2017 rainfall data in Dhunche, seven kilometres southwest to Syabru-Bensi (green), and Timure, nine kilometres to the north (purple) (courtesy of Nepal Department of Meteorology, Kathmandu). Pre-earthquake water temperature and DIC concentration of SBP0 are compiled from our database and the literature[30]. Timing of the Gorkha earthquake, the Chilime hot spring cessation and the Sanjen $CO_2$ outburst are displayed as vertical grey lines. Possible pre-seismic water-cooling of 0.6 ± 0.2 °C is observed 2 weeks before the earthquake, followed by a progressive warming reaching 4.4 ± 0.2 °C about 1 year after the earthquake, which persisted >2 years after the mainshock. Water temperature returned to pre-earthquake values in January 2018, more than 2.7 years after. Water-cooling was observed a few weeks before the earthquake at three other hot springs of the Upper Trisuli valley (Supplementary Fig. 9). The DIC concentration increased by 29 ± 2% after the mainshock and returned to pre-earthquake values in September 2017. Compared with $CO_2$ flux values obtained in 2006 (ref.[33]), close to local background level, the post-seismic values were significantly larger in November 2015 and January 2016 (increase factors of 150 ± 30 and 160 ± 20 for K+6 and K+12, respectively). The $CO_2$ fluxes peaked in November 2016 and continued to show higher values in January 2018 compared with 2006

Carbon isotopic ratio of the $CO_2$ emissions ($\delta^{13}C$; see Methods) was systematically measured at all occurrences (Table 1 and Supplementary Table 1), giving the most comprehensive data set ($n = 77$) so far in a seismically active area without magmatic activity. Along the rupture zone, average $\delta^{13}C$ values range from −6.9 ‰ to −0.1 ‰, and appear similar at the various sites of a given valley. $CO_2$ emissions in Sanjen and Syabru-Bensi, although separated by eight kilometres, have comparable $CO_2$ concentration (96–98%) and $\delta^{13}C$ signature (from −0.9 ‰ to −0.7 ‰), precluding shallow sources, but instead suggesting similar $CO_2$ source and transport from a crustal-scale reservoir. In addition, the $\delta^{13}C$ values remain relatively stable after the earthquake (Supplementary Fig. 4), indicating that the earthquake revealed a pre-existing $CO_2$ reservoir.

**Hydrothermal unrest triggered by the Gorkha earthquake**. The $CO_2$ emissions were associated with changes of hydrothermal activity (Table 2 and Supplementary Table 2). In Syabru-Bensi,

new hot springs appeared after the earthquake, with a few persisting in January 2018, > 2.7 years after the mainshock. The temperature of the main Syabru-Bensi hot spring (SBP0), which had been stable at 60.7 ± 0.1 °C for more than 12 years before the earthquake (Fig. 3), increased to 64.1 ± 0.3 °C after the earthquake (November 2015 – January 2016), along with a significant flow rate increase of 16 ± 1%. Water temperature peaked about 1 year after the mainshock. Water warming of SBP0 persisted more than 2 years after the mainshock, before starting to decrease to pre-earthquake values in January 2018, more than 2.7 years after. Other springs in Syabru-Bensi (SBB5) also showed warming, persisting in January 2018 (Supplementary Fig. 9). A slight pre-seismic water-cooling was possibly detected in Syabru-Bensi (Fig. 3). More significantly, three other hot springs of the same valley (Timure, Langtang and Chilime) also showed cooling a few weeks before the earthquake (Supplementary Fig. 9). In Tatopani (Budhi Gandaki valley), locals reported hot spring temperature decrease a few weeks before the earthquake. While co-, post- and

**Table 2 Post-seismic effects on the hot springs at nine hydrothermal sites in Central Nepal. Data of temperature, flow rate, and dissolved inorganic carbon concentration and isotopic ratio of hot springs are compiled before and after the Gorkha earthquake**

| Site | Location | Name | Type | Before/after Gorkha earthquake | Spring temperature (°C) | Spring flow rate (L s⁻¹) | Dissolved inorganic carbon (DIC) | | | Post-seismic effect |
|---|---|---|---|---|---|---|---|---|---|---|
| | | | | | | | $N_{meas}$ | $C_{DIC}$ (mmol L⁻¹) | $\delta^{13}C_{DIC}$ (‰) (V-PDB) | |
| *Budhi Gandaki Valley* | | | | | | | | | | |
| Machhakhola | Eastern Bank | BUD0 | HS[a], BB, DDS | After | 59.8 ± 0.2 | n.m. | 1 | 26.4 ± 0.3 | –0.5 ± 0.2 | Increase (New) |
| Tatopani | Southern Secondary Springs | BUD3B | HS[a] | After | 43.2 ± 0.1 | n.m. | 1 | 25.6 ± 0.9 | 0.9 ± 0.3 | Increase (New) |
| | Main Springs | BUD4B | HS | Before[b] | 30 ± 1 | n.m. | 1 | 12.2 ± 1.2 | n.m. | Decrease (Temp.) |
| | | | | After | 22.2 ± 1.8 | 0.061 ± 0.002 | 1 | 15.8 ± 0.3 | 3.7 ± 0.3 | |
| | | BUD4C | HS | Before[b] | 50 ± 1 | n.m. | 1 | 19.1 ± 1.9 | 3.3 ± 0.3 | Decrease (Temp.) |
| | | | | After | 48.5 ± 1.3 | 0.19 ± 0.04 | 1 | 22.0 ± 0.2 | 1.6 ± 0.3 | |
| *Upper Trisuli Valley* | | | | | | | | | | |
| Sanjen | Tunnel | TSJ3 | HS[a], DDS | After | 20.3 ± 0.1 | n.m. | 1 | >55 | –9.1 ± 0.7 | Increase (New) |
| | Piezometer | DH1 | AQ, DDS | After | 18.9 ± 0.6 | n.m. | 2 | 43.8 ± 0.1 | 1.6 ± 0.1 | Unknown |
| | Piezometer | DH2 | AQ, DDS | After | 19.1 ± 0.8 | n.m. | 1 | 43.3 ± 0.1 | 2.1 ± 0.1 | Unknown |
| Chilime | Hot Spring | CHI | HS, DDS | Before[b,c,d,e,f,g] | 48.9 ± 0.4 | 5.0 ± 0.2 | 3 | 13.8 ± 1.3 | 8.3 ± 0.4 | Decrease (Cessation) |
| | | | | After | No spring | 0 | 0 | | | |
| Syabru-Bensi | Western Bank Secondary Springs | GZ3 | HS, DDS | Before | 20.3 ± 1.2 | n.m. | 1 | 1.9 ± 0.1 | n.m. | Increase (Temp.) |
| | | | | After | 23.4 ± 0.1 | n.m. | 1 | 3.2 ± 0.1 | 3.7 ± 0.1 | |
| | | FF2 | HS[a], BB | After | In Trisuli River | n.m. | 0 | n.m. | n.m. | Increase (New) |
| | Main Springs | SBP0 | HS, DDS | Before[b,c,d,g] | 60.7 ± 0.1 | 0.087 ± 0.004 | 4 | 25.8 ± 1.6 | 4.7 ± 0.7 | Increase (Temp., flow, $C_{DIC}$, $\delta^{13}C_{DIC}$) |
| | | | | After | 64.1 ± 0.3 | 0.104 ± 0.007 | 5 | 34.3 ± 1.5 | 1.0 ± 0.1 | |
| | | SBB5 | HS, DDS | Before[g] | 31.8 ± 0.3 | 0.282 ± 0.009 | 2 | 25.8 ± 3.6 | 0.9 ± 0.1 | Increase (Temp., flow) |
| | | | | After | 34.9 ± 0.2 | 0.37 ± 0.02 | 4 | 29.6 ± 0.9 | 0.2 ± 0.3 | |
| | | SBC2 | HS, DDS | Before[g] | 50.1 ± 1.9 | n.m. | 2 | 17.8 ± 1.1 | 2.6 ± 0.1 | Decrease (Temp.) |
| | | | | After | 40.2 ± 0.5 | n.m. | 1 | 22.3 ± 0.1 | 2.1 ± 0.1 | |
| | | SBM | HS[a], DDS | After | 34.9 ± 0.2 | n.m. | 1 | 30.3 ± 0.1 | –0.6 ± 0.1 | Increase (New) |
| | | SBN | HS[a], DDS | After | 37.3 ± 0.8 | n.m. | 1 | 27.4 ± 0.1 | 2.3 ± 0.1 | Increase (New) |
| | | SBN2 | HS[a], DDS | After | 39.2 ± 0.6 | 0.50 ± 0.02 | 3 | 17.3 ± 1.1 | 2.0 ± 0.2 | Increase (New) |
| North Syabru | Western Bank | TT1 | HS, BB | Before[b,c] | 24.38 ± 0.07 | n.m. | 4 | 35.8 ± 2.9 | 12.3 ± 0.7 | Increase (Temp.) |
| | | | | After | 25.3 ± 0.1 | n.m. | 2 | 38.3 ± 2.3 | –0.6 ± 0.1 | |
| Timure | | TIM | HS, DDS | Before[b,c,f] | 61.3 ± 2.2 | 0.20 ± 0.02 | 5 | 17.3 ± 1.0 | 3.3 ± 0.1 | Increase (Temp.) |
| | | | | After | 70.7 ± 1.0 | n.m. | 2 | 14.2 ± 0.2 | 2.1 ± 0.1 | |
| Langtang | | LPAH | HS | Before[c,e] | 41.0 ± 0.1 | n.m. | 2 | 5.1 ± 0.2 | –4.8 ± 0.1 | Decrease (Temp.) |
| | | | | After | 37.3 ± 0.5 | n.m. | 1 | 4.8 ± 0.2 | –3.2 ± 0.2 | |
| *Bhote Koshi Valley* | | | | | | | | | | |
| Kodari | | KOD | HS | Before[b,g] | 44.6 ± 1.2 | 3.0 ± 0.2 | 1 | 8.4 ± 0.3 | –8.7 ± 0.1 | Increase (Temp.) |
| | | | | After | 50.7 ± 0.1 | 3.00 ± 0.04 | 0 | n.m. | n.m. | |
| | | KOD2 | HS[a] | After | 48.1 ± 0.1 | n.m. | 0 | n.m. | n.m. | Increase (New) |

n.m.: not measured, HS: hot spring, BB: bubbles, DDS: diffuse degassing structure, AQ: aquifer degassing
[a]New spring that appeared after the Gorkha earthquake
Compilation of new original data and of data compiled in ref.[30]., and in particular from: [b]ref.[32]; [c]refs.[31] and [52]; [d]ref.[63]; [e]ref.[64]; [f]ref.[51]; [g]ref.[33]

sometimes pre- seismic changes in hot spring temperature have been reported in the literature[16,19], a two-year-long or longer warming is unusual.

The most spectacular change in hydrothermal activity, however, is the complete cessation of the Chilime hot spring at the end of October 2015, after periods of unusual intermittence between April and June 2015 (Supplementary Fig. 10). Before the earthquake, this spring had a stable flow rate of > 5 L s⁻¹; it was the pillar of local economy, being the most important in Central Nepal after the Kodari hot spring (Fig. 1). The village elders had previously reported spring intermittence at the time of the 1934 earthquake, but no cessation. Particularly impressive is the fact that it happened at about the time of the $CO_2$ outburst in Sanjen.

Dissolved Inorganic Carbon (DIC) concentration and isotopic ratio ($C_{DIC}$ and $\delta^{13}C_{DIC}$; see Methods) were systematically measured for all hot spring waters (Table 2 and Supplementary Table 2). In Syabru-Bensi, where the number of measurements

is significant ($n = 8$), $C_{DIC}$ of the SBP0 hot spring increased by 29 ± 2% after the mainshock and, in September 2017, returned to values measured before the mainshock (Fig. 3), while Ca and Na concentrations remained similar within 5%. The $\delta^{13}C_{DIC}$ decreased also significantly (Supplementary Fig. 11). These observations suggest a larger amount of dissolved carbon in the SBP0 hot spring after the earthquake, which is compatible with the aforementioned increase in gaseous $CO_2$ emissions. Available pH values (Supplementary Fig. 9) also show a slow return to pre-earthquake values, with anomalously high values before the earthquake, a fact to be interpreted with caution given the lack of additional geochemical information. In Syabru-Bensi, dissolved radon and radium concentrations in SBP0 and SBB5 hot springs changed after the earthquake (Supplementary Fig. 9). In January 2018, >2.7 years after the mainshock, the warmest spring in Syabru-Bensi (SBP0) returned to pre-earthquake conditions, while those more dependent on superficial effects (e.g., SBB5) did not yet.

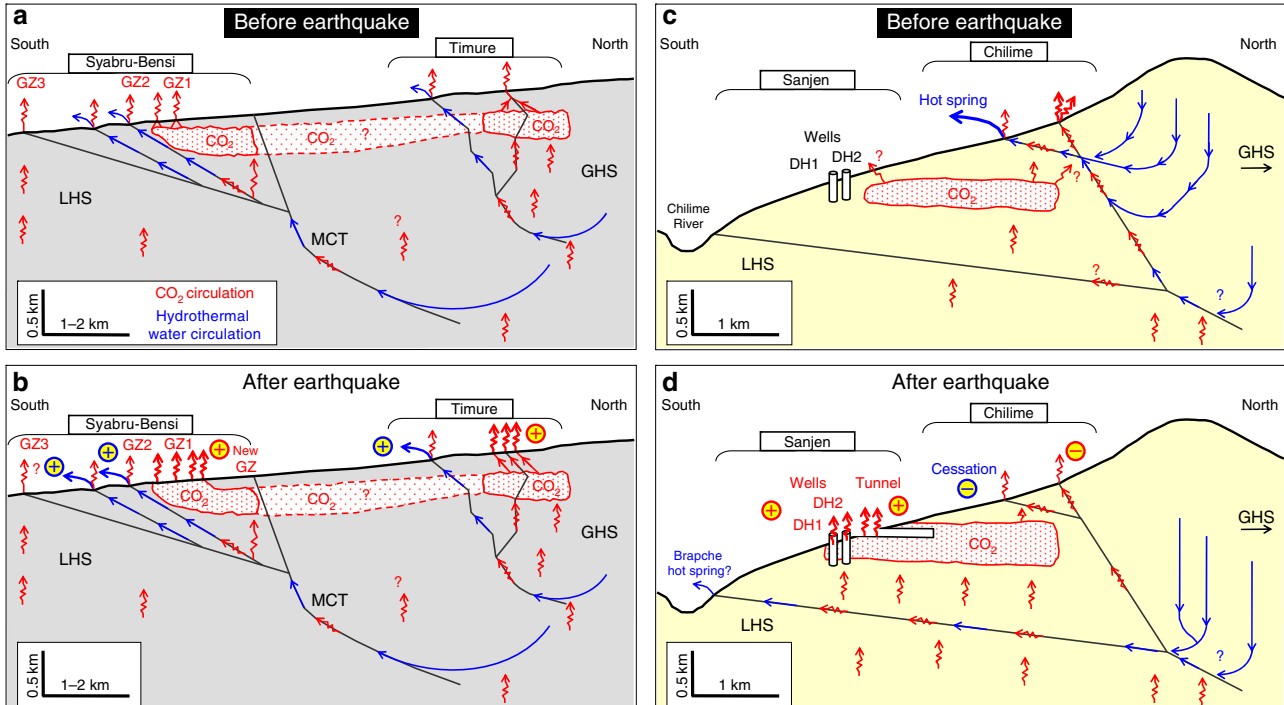

**Fig. 4** Conceptual models of the carbon dioxide and hydrothermal transport before and after the Gorkha earthquake. Pre- and post- seismic models are shown separately for **a–b** the Bhote Kosi valley (Syabru-Bensi and Timure) and **c–d** the Chilime valley (Chilime, Sanjen and Brapche). The $CO_2$ is produced at depth and a significant fraction is transported to the surface by hydrothermal circulations, which efficiently take advantage of fault network; some $CO_2$ can also move upwards independently, reaching the surface without hot spring (Sanjen, Syabru-Bensi GZ3) and accumulate in the subsurface reservoir revealed by $CO_2$ emissions following the earthquake. The main effect of the earthquake is an increase of vertical permeability with transient or permanent changes

**First non-volcanic earthquake-induced gaseous changes.** Hydrologic responses to seismic stimulation have been observed in numerous instances[10], and correlated with Seismic Energy Density (SED) or Peak Ground Velocity (PGV) values. In Lassen (Cascades Range, USA) for example, a 2014 volcano-seismic swarm peaking at $M_w = 3.85$ at 5.7 kilometres distance, corresponding to SED~$0.2\,\mathrm{J\,m^{-3}}$ and PGV~$0.2\,\mathrm{cm\,s^{-1}}$, caused an outburst of geothermal fluids, explained by a two-fold permeability increase[22]. In the case of the Matsushiro earthquake swarm in Japan, modelling indicates the necessity of a 2-order-of-magnitude permeability increase with a small overpressure of only a few megapascals[9]. In Syabru-Bensi, the Gorkha earthquake produced SED~$63\,\mathrm{J\,m^{-3}}$ and PGV~$26\,\mathrm{cm\,s^{-1}}$ (see Methods; Supplementary Table 3), hence strong enough to affect the hydrothermal system, with several aftershocks maintaining high ground motion during the following 31 months (Supplementary Fig. 12). These estimates are confirmed by GPS time-series of the Chilime station[44,45] (Fig. 1), giving PGV~$49\,\mathrm{cm\,s^{-1}}$ (Supplementary Fig. 13). While the first clearly documented instances in the Himalayas, our observed near-field $CO_2$ outbursts and hydrothermal unrest remain compatible with previously compiled earthquake-induced changes (Supplementary Fig. 14). Our observed changes in the $CO_2$ emissions are however the first earthquake-induced gaseous changes in a non-volcanic region.

## Discussion

The observations following the Gorkha earthquake give some indications that the standard model, where $CO_2$ is degassed from hydrothermal waters, needs some modifications at least in part. Indeed, previous work in Syabru-Bensi[34] identified large $CO_2$ emissions without the presence of an important hot spring in the vicinity. Now, the evidence for an extended reservoir of crustal $CO_2$ is overwhelming, which we accommodate with the following conceptual models (Fig. 4). In the case of the Trisuli valley (Syabru-Bensi and Timure) (Fig. 4a, b), hydrothermal circulations are important and increased after the earthquake. The main $CO_2$ source can therefore be the hydrothermal circulations, with near surface degassing and subsurface accumulation in the whole fault zone. Possible additional $CO_2$ can also be released in the footwall directly from the production source. The earthquake caused (Fig. 4b) an increasing water discharge, with increased $CO_2$ degassing, or better communication to the surface of previously accumulated $CO_2$. In the case of the Chilime valley (Chilime, Sanjen and Brapche) (Fig. 4c, d), however, the pervading presence of $CO_2$, revealed by the earthquake, kilometres away from hot springs, is better explained if $CO_2$ is directly accumulated from below, possibly through the water table, without advection by hydrothermal circulations.

Strikingly, $CO_2$ emissions with similar isotopic anomaly were observed over the whole region, whenever faults, boreholes or a tunnel gave the opportunity. These observations also attest the presence of a large, relatively shallow, reservoir of $CO_2$ in the Himalayan crust, suggesting that metamorphic $CO_2$ produced at depth is huge, as independently shown by petrological estimates[47,48], and unlikely sequestered. Nevertheless, as hot water could also be found when deep boreholes are available, the overwhelming presence of hot water could match the evidence of an extended $CO_2$ reservoir, and the debate cannot be considered as closed.

The effects of the earthquake on our hydrothermal systems are not unexpected. Indeed, our observed $CO_2$ emissions appear consistent with hydrothermal outburst effects, as observed in Lassen[22], suggesting that apparent two-fold permeability increases and/or changes in hydrothermal pathways for pre-existing $CO_2$ emissions could have been major effects of the Gorkha

earthquake. First-order modelling indeed confirms that at least a doubling of permeability could explain $CO_2$ flux increases (Supplementary Fig. 15). In addition, first-order modelling supports that a lasting permeability increase[18] of 10–20% could accommodate the Syabru-Bensi hot spring warming and increased flow (Supplementary Figs. 16 and 17). These vertical permeability increases appear compatible with post-seismic afterslip[45]. The global pre-seismic water-cooling could be evidence for a pre-seismic dilatancy effect[49] of the impending rupture zone.

Compared with reported earthquake-induced hydrological effects[10,16,19,46], some of our observations are unusual: first, the six-month delay of the Sanjen $CO_2$ outburst and the shut down of the Chilime hot spring, and second, the persistence (and rising at some places) > 2.7 years after in January 2018 of all the increased $CO_2$ emissions and hydrothermal unrest. In Lassen[22], the permeability increase was inferred to be maintained for 50–60 days. In the Napa valley[13], a vertical permeability change, following diffusion of a pressure pulse, was proposed to explain post-seismic stream discharge, suggesting basin-scale fluid diffusivity of 0.1 $m^2 s^{-1}$. Taking this conservative value, for one-kilometre spatial scale, the timescale for fluid displacement is ~ 120 days, compatible with our six-month delay time. This suggests that the Sanjen $CO_2$ outburst and the Chilime cessation likely result from hydrogeological fluid diffusion.

Other explanations are, however, possible. In Central Nepal, post-seismic deformation (afterslip on the MHT) and associated aftershocks remained active several months after the mainshock. About 6 months after the mainshock, contemporaneous with the Sanjen outburst and the Chilime cessation (Fig. 3a), the number of seismic events producing PGV>1 cm $s^{-1}$ in Syabru-Bensi increased, and a linear trend in the GPS time-series, continuing to October 2017 about 2.5 years after the mainshock, was initiated. Thus, some post-seismic $CO_2$ emissions and hydrothermal unrest may be related to changes in the state of post-seismic relaxation. Incidentally, the time of the Sanjen outburst and the Chilime cessation coincides with renewal of aftershock activity near Syabru-Bensi in the fall of 2015 (Fig. 3a), including an $M_L = 5.3$ event a few weeks before. In addition, two $M_L > 4$ events occurred at about the same time (end of October – beginning of November), less than 15 kilometres from Syabru-Bensi, producing PGV > 1 cm $s^{-1}$ (Supplementary Table 3).

The Himalayan hydrothermal systems appear highly sensitive to small deformation rates and are therefore in near-critical condition. This suggests that post-seismic relaxation of co-seismic stress may result from pore pressure changes, and that metamorphic $CO_2$ may in turn play a role in the installation of the next inter-seismic regime. Alternatively, the still evolving $CO_2$ emissions in Syabru-Bensi, Timure and Chilime may also indicate a currently unstabilized system, able to diverge unpredictably. Given the issue of a pending mega-earthquake in the region[42], long-term monitoring of $CO_2$ emissions should seriously be considered as a chance to capture possible pre-earthquake signals.

In this paper, we have presented the first assessment of $CO_2$ emissions triggered by a major earthquake, demonstrating the coupling between mechanical deformation and fluid transport properties at the crustal scale, highlighting that crustal deformation dynamically affects permeability during the seismic cycle. The large post-seismic $CO_2$ emissions observed in the Narayani basin suggest non-stationary metamorphic $CO_2$ production, and that its current estimate[30,32] (>$1.3 \times 10^{10}$ mol year$^{-1}$ with >$(1.0 \pm 0.2) \times 10^8$ mol year$^{-1}$ from direct gaseous $CO_2$ emissions), independently confirmed by petrological studies[48], may be enhanced during a significant fraction of the seismic cycle. Metamorphic $CO_2$ and its transport therefore emerge as an essential component of mountain build-up and the associated dynamics of large earthquakes.

## Methods

**Carbon dioxide flux measurement and mapping.** The accumulation chamber method[50] was used to measure surface $CO_2$ flux and to quantify the total $CO_2$ discharge of a given site and the associated uncertainties, whose assessment is based on numerous systematic tests and our 10-year experience[34]. The method is robust, even in remote locations[51] and during the monsoon[36], and allows the measurement of $CO_2$ fluxes over more than five orders of magnitude (Supplementary Fig. 2). The increase in $CO_2$ concentration in the chamber is measured using various portable infrared $CO_2$ sensors (Testo™ 535, Testo AG, Germany; Airwatch™ PM 1500, Geotechnical Instruments Ltd., UK; Vaisala™ CARBOCAP® Hand-Held GM70, Finland), that are regularly inter-calibrated in the laboratory. The $CO_2$ flux is expressed in grams per squared metre per day (g m$^{-2}$ d$^{-1}$). The total $CO_2$ discharge, expressed in mol s$^{-1}$ (or ton d$^{-1}$), is estimated using the $CO_2$ flux data-set by kriging and interpolation procedures[34]. $CO_2$ fluxes ($n = 1720$) and total $CO_2$ discharges obtained before the Gorkha earthquake (from 2006 to 2011) are published elsewhere[30,33,34,51]. Here we present for the first time $CO_2$ fluxes ($n = 1668$) and total discharges obtained after the earthquake in the Marsyandi, Budhi Gandaki and Upper Trisuli valleys relying on seven measurement campaigns carried out in November 2015, in January, May and November 2016, in January and September 2017, and in January 2018. Pre-earthquake and post-earthquake campaigns were performed outside the monsoon periods to reduce the meteorological effects on $CO_2$ flux data. Every uncertainty is given around one-sigma standard deviation (68% confidence level) and averages are arithmetic means except otherwise stated. Data are summarised in Table 1 and Supplementary Table 1.

**Carbon dioxide flux measurement through a water layer.** The bubbling $CO_2$ flux from or through water was measured at the new degassing site of Machhakhola (Budhi Gandaki valley) in the following manner. A collecting container was installed upside down on the water with a pipe leading to an accumulation chamber installed on the ground nearby (Supplementary Fig. 8). Then, the flux was measured in this accumulation chamber as described above. This method yielded a minimum value to the large $CO_2$ discharge observed in Machhakhola (Supplementary Movie 2).

**Carbon isotopic composition of the gas phase.** We sampled gas in the field using evacuated glass tubes. $CO_2$ fraction of the gas sample was determined manometrically. The $\delta^{13}C$ of $CO_2$ of the gas, expressed in per mil relative to the standard values of Vienna Pee Dee Belemnite (V-PDB), was measured after off-line purification by mass spectrometry on a Finnigan™ MAT-253 mass spectrometer (Thermo Electron Corp., Germany) in CRPG (Nancy, France)[32,34]. External repeatability of a given sample was ±0.1 ‰. The twenty $\delta^{13}C$ values measured before the Gorkha earthquake are published elsewhere[30,33,34,52]. Fifty-seven measurements were carried out after the earthquake from January 2016 to January 2018. Data are summarised in Table 1 and Supplementary Table 1.

**Water temperature, pH and flow rate measurements.** To measure water temperature of springs, we used thermometers (Generic TP101 Digital Thermometer, China) regularly inter-calibrated in the laboratory with a reference thermometer not currently used in the field (Digital Thermometer model 4400 Ertco Eutechnics, USA), and compared with high-precision ($10^{-3}$ °C) and high-sensitivity ($10^{-4}$ °C) thermometers (Seabird™ 39plus, Sea-Bird Scientific, USA). Due to the time response of the instruments, several minutes are needed to measure temperature. Experimental uncertainty of a given measurement was ±0.1 °C. The pH of the water in thermal springs was measured with various pH metres (H170 Portable pH metre, Hach, Germany; HI98107 and HI98130 pH metres, Hanna Instruments, USA), systematically recalibrated in the field using buffer solutions. Experimental uncertainty of a given measurement was ±0.1. The flow rate of thermal springs was determined using stopwatch and measured cylinders or buckets and was repeated at least three times. Data are summarised in Table 2 and Supplementary Table 2.

**Carbon isotopic composition and DIC concentration in water.** We sampled every water in the field using two to three glass screw cap vials of 12 millilitres volume each. DIC concentration ($C_{DIC} = [H_2CO_3] + [HCO_3^-] + [CO_3^{2-}]$) and its isotopic composition ($\delta^{13}C_{DIC}$), expressed in mmol L$^{-1}$ and in per mil relative to V-PDB, respectively, were determined using a gas chromatograph coupled to an isotope ratio mass spectrometer (GCIRMS, GV 2003, GV Instruments, UK) in IPGP (Paris, France). The whole procedure is described elsewhere[53]. The relative experimental uncertainty of $C_{DIC}$ was 1–2% and the experimental uncertainty of a given $\delta^{13}C_{DIC}$ measurement was ±0.1‰. For a given sample, final values correspond to weighted arithmetic averages of two to three measurements. The $C_{DIC}$ and $\delta^{13}C_{DIC}$ values measured before the Gorkha earthquake are published elsewhere[30–33]. A total of 71 measurements were carried out after the earthquake from January 2016 to January 2018. Data are summarised in Table 2 and Supplementary Table 2.

**Water radon-222 and radium-226 concentration measurements.** Dissolved radon concentration in water was measured in the field by emanometry in air[54]. After air–water equilibrium is reached by manual shaking, radon concentration is inferred from scintillation flask sampling and photomultiplier counting, as described elsewhere[51]. Radon concentration in water is expressed in Bq L$^{-1}$.

Experimental uncertainty ranged from 5 to 30%. The radium concentration in water was similarly measured in the IPGP laboratory after keeping the bottle closed for at least 50–80 days[55]. Expressed in mBq L$^{-1}$, the experimental uncertainty was the same as for radon concentration in water. Here we present only the dissolved radon and radium concentrations in the main Syabru-Bensi hot springs (Supplementary Fig. 9).

**Detection of thermal springs and $CO_2$ degassing areas**. The $CO_2$ degassing areas were detected based on pervasive hydrogen sulphide odour, measurement of high radon-222 flux (radioactive gas of half-life 3.8 days), surface temperature anomalies, occurrence of water bubbles, presence of cavities, occurrence of inactive or active travertine deposits, and discussion with local people, or a combination of the above[30,51]. In remote places, the detection of previously unknown thermal springs relied on the use of hand-held thermal infrared cameras[34,35,51] (model 880-V3 before 2015 and model 875-1i after, Testo™ AG, Germany).

**Determination of SED, PGV and PDS**. At the sites which experienced the most significant post-seismic changes in $CO_2$ emissions and hot springs, we calculated SED[56] and vertical PGV[57] produced by the Gorkha earthquake and its main aftershocks. Empirical equations are used to estimate SED[56] and PGV[58]. Peak Dynamic Stress (PDS) is estimated using: PGV × shear modulus ($3 \times 10^{10}$ Pa) / shear wave velocity (3500 km s$^{-1}$). For the mainshock and six main aftershocks, values of SED, PGV and PDS at eight hydrothermal sites are gathered in Supplementary Table 3. In addition, SED, PGV and PDS produced at sites of the Upper Trisuli valley by three aftershocks located near Syabru-Bensi are also given.

**Determination of Seismic Moment Release Rate**. We calculated the Seismic Moment Release Rate (SMRR) produced by the aftershocks of the Gorkha earthquake using the NSC earthquake catalogue from 25 April 2015 to 31 December 2017. We converted local magnitude ($M_L$) given by the catalogue into moment magnitude ($M_w$), using the following linear calibration: $M_w = 1.109 M_L - 0.626$. The seismic moment was obtained using the classical relationship: $\log(M_0) = 1.5 M_w + 9.1$. The SMRR, expressed in N.m per 20 days, was calculated for all seismic events around Syabru-Bensi and is shown in Fig. 3.

**Modelling of earthquake-induced changes**. To study the effect of permeability changes on $CO_2$ flux before and after the earthquake, we used a 2-D model, described elsewhere[37], in which a vertical fault (f) surrounded by two media (a and b) transports the gaseous $CO_2$ to the surface by advection. Each medium is characterised by connected porosity ($\varepsilon$) and permeability ($k$). A pressure source is fixed at depth. Darcy's law is defined according to a pressure distribution ($p$) that follows $\Delta p^2 = 0$ (ref.[59]). An example calculation is shown in Supplementary Fig. 15.

To study the effect of permeability changes on spring water flow rate at the surface, we adapted the 2-D fault model described above[37], to the calculation of water flow rate. We consider a vertical fault (f) driving the water flow to the surface, surrounded by two media (a and b). Each medium is characterised by permeability ($k$). Pressure source is fixed at depth. Solutions of the pressure distribution are expressed as a sum of exponential terms on horizontal axis, modulated by a sinusoidal signal along vertical axis[37]. An example of calculation is shown in Supplementary Fig. 16.

To study the effect of water flow rate changes on hot spring exit temperature, we relied on an analytical first-order model[60] (Supplementary Fig. 17a). We consider a vertical conduit, characterised by perimeter $p$ and thermal conductivity $K_T$, that drives hot water from depth ($z = h$) to the surface ($z = 0$), with flow rate $Q$, density $\rho$ and specific heat $C_P$. Temperature at depth $h$ is defined by $T(h) = \beta h + T(0)$, where $\beta$ is the thermal gradient, which is poorly constrained in this hydrothermal region[61] and taken equal to 55–75 °C km$^{-1}$. We consider a quasi-static state having characteristic length $\lambda_T$ and we define $\theta = T - T(0)$ and $\alpha = \rho C_P Q \lambda_T / (pK_T)$. Using the initial conditions, the differential equation $\alpha h (d\theta/dz) + \theta = -\beta z$ has the following solution: $\theta = \alpha h \beta (1 - e^{-1/\alpha})$. Examples of calculation of the water temperature as a function of water flow rate, for two values of the thermal gradient, are shown in Supplementary Fig. 17b.

**Data availability**. The data that support the findings of this study are available in the article, in Supplementary Information, and from the corresponding author upon request (girault@ipgp.fr).

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

## Acknowledgements

We warmly thank Soma Nath Sapkota and the Department of Mines and Geology for enthusiastic support over the years. We are indebted to Buddha Lama, Niraj Jairu and Ohm Tharum for assistance in the field, and to Nabin Tamang and Dawa Tamang for giving access to their property in Syabru-Bensi. Svetlana Byrdina is thanked for preparing the $CO_2$ flux data obtained in 2006–2007. We are thankful to Jean-Philippe Avouac, Steve Ingebritsen, Yves Gaudemer, Damien Jougnot, Christian Fouillac, Pascal Bernard, Hélène Lyon-Caen and Marc Chaussidon for fruitful discussions. Thomas Rigaudier is thanked for the measurement of carbon isotopic ratios in CRPG. Carine Chaduteau is thanked for the measurement of carbon isotopic ratios in IPGP. Narayan P. Malla is thanked for giving access to the Rasuwagadhi Hydroelectric Project. Hélène Bouquerel, François Prevot, Marie Bouih and Roxane Ferry are thanked for their participation in the fieldwork. Financial support for this work was essentially provided by public and private funding of the IPGP team "Physics of Natural Sites", partially complemented in 2015 by a TelluS-SYSTER INSU CNRS funding to F.G. and in 2016 by a CNRS attribution to C.F.-L. This is IPGP contribution number 3942.

## Author contributions

L.B.A. provided aftershock catalogue and processed Fig. 1. C.F.-L. performed carbon isotopic measurements of the gas phase in the laboratory. P.A. performed carbon isotopic and dissolved inorganic concentration measurements of the water phase in the laboratory. B.P.K. and M.B. participated to field $CO_2$ flux measurements. S.S.M. participated in the discussions and managed the access to the Sanjen Hydroelectric Project sites. C.G. and F.R. performed water measurements a few weeks before the Gorkha earthquake at four hot springs. L.B. managed the assistance to NSC, seismic data processing, and provided detailed DEM and shared experience. F.G. performed water measurements, $CO_2$ and radon flux measurements, and collected all samples. F.P. performed radon flux measurements and processed Supplementary Fig. 5. F.G. and F.P. wrote the paper.

## Additional information

**Competing interests:** The authors declare no competing interests.

