## [Peer Review File · Nature Communications]

Reviewers' comments:

Reviewer #1 (Remarks to the Author):

The ms is very interesting because it for the first time reports an anomalous metamorphic CO₂ degassing in association with a large earthquake in the Himalaya region. However in my opinion there are some (main) problems that should be fixed before an eventual publication in NC.

1) I have some concerns about the use of the term 'eruption' (sometime 'massive CO₂ eruptions'!) referring to the phenomenology described in the ms. The described processes are very interesting especially for the geodynamic contest where they occur, but quantitatively are of minor relevance in comparison with other CO₂ emissions of the Earth. For example I note that the total CO₂ emitted by all the gas emissions in the region (estimated to be > 1x10⁸ mol/y, i.e. a few thousands tons of CO₂ per year) is generally much lower than the flux observed in other not volcanic areas of the Earth (even from single vents) (see, e.g., the data reported in Burton et al., 2013 or in Moerner and Etiope, 2002). For example in Italy a single emission of cold CO₂ from a not volcanic area discharges about 2000 tons of CO₂ per day (Chiodini et al., 2010), i.e. about 160 times higher than the estimated CO₂ flux from all the gas emissions in Nepal. In conclusion I suggest the authors to use a different terminology (i.e. emission instead of eruption) and to don't use the term "massive" or similar

2) The authors does not give reliable proves that the increase in the degassing observed after the 2015 earthquake can not be due, at least partially, to different meteorological conditions that can heavily affect the soil diffuse degassing. The authors state "... These emissions showed remarkable temporal stability, even during monsoon17,..." The cited reference refers to a work that regards just one site and that, at least from the title, seems to deal with Radon instead of CO₂ (17: Richon, P. et al. Temporal signatures of advective versus diffusive radon transport at a geothermal zone in Central Nepal. J. Environ. Radioact. 102, 88-102 (2011)). It is worth to note that important seasonal (and daily) variations of intensity similar to the variation described in the work (e.g. factor 1.6) are normally observed in the sites where soil CO₂ fluxes are continuously measured (see for example Fig. 3 in Granieri et al. 2003 and Fig. 2 in Viveiros et al. 2014). Similar variations, that were observed in volcanic environments, could affect also the emission in Nepal because the variations do not depend on the origin of the gas but on meteorological conditions (i.e. atmospheric pressure changes, soil humidity, soil temperature etc.). I think that the authors should discuss how, also in the absence of continuous measurements, they can infer the deep origin of the observed degassing increases (for example because the contemporary occurrence in different location of the increase of CO₂ degassing after the earthquake, or because in some sites the measurements were done in the same period of the year, etc.)

3) The third problem regards the absence of a conceptual model explaining the occurrence of CO₂ emissions in association with thermal springs (see Fig. 1). The physical models proposed by the authors treat the two processes (CO₂ emission and thermal water discharges) separately but I think that a more realistic scenario should consider that the gas is separated by the thermal waters during their rise (and depressurization) from depth to the surface. What is the gas content of the thermal waters? What is the chemical and isotopic compositions of the gas and of the thermal waters? What changes were observed in the chemical and isotopic compositions of gas and waters?

4) The variation in the gas flux are reported in a table (extended data table 1) where the changes are not easily readable. I suggest to report the variation (before and after) also in the map of Fig. 1 where the authors could use different symbols for CO₂ emissions, CO₂+thermal water and thermal waters and different colours to indicate the variation (e.g., red where the fluxes increased, blue where are stable, green where the fluxes decreased, gray where no comparative data are available)

5) References. I suggest the authors to read the review article "The Role of Fluids in Tectonic and Earthquake Processes" (Miller 2013) and in particular the chapter 4.5 "Documented Fluid-Driven Earthquake Sequences" where are reported a series of cases of fluids anomalies during earthquakes and to add relevant papers to the reference list of the ms.

In conclusion I think that the ms is interesting but that in the present form can not be published. However, I think also that the authors could fix the above problems.

Cited references

- Burton, M. R., Sawyer, G. M. & Granieri, D. Deep carbon emissions from volcanoes. *Rev. Mineral. Geochem.* 75, 323–354 (2013).
- Chiodini G, Granieri D, Avino R, Caliro S, Costa A, Minopoli C, Vilardo G (2010) Non-volcanic CO₂ Earth degassing: Case of Mefite d'Ansanto (southern Apennines), Italy. *Geophys Res Lett* 37:L11303, doi: 10.1029/2010GL042858
- Granieri, D., G. Chiodini, W. Marzocchi, and R. Avino (2003), Continuous monitoring of CO₂ soil diffuse degassing at Phlegraean Fields (Italy): Influence of environmental and volcanic parameters, *Earth Planet. Sci. Lett.*, 212, 167–179.
- Miller (2013), The Role of Fluids in Tectonic and Earthquake Processes. *Advances in Geophysics* 54:1-46 • December 2013
- Mörner NA, Etiope G (2002) Carbon degassing from the lithosphere. *Global Planet Change* 33:185-203
- Viveiros, F., J. Vandemeulebrouck, A. P. Rinaldi, T. Ferreira, C. Silva, and J. V. Cruz (2014), Periodic behavior of soil CO₂ emissions in diffuse degassing areas of the Azores archipelago: Application to seismovolcanic monitoring, *J. Geophys. Res. Solid Earth*, 119, 7578–7597, doi:10.1002/2014JB011118.

Reviewer #2 (Remarks to the Author):

This is an interesting paper that presents a simple but important set of observations. There are a number of minor corrections needed to the manuscript which I have marked on the attached version*.

The explanation for the change in spring temperature in the methods section needs to be better explained.

*Editorial Note: These comments, excluding textual revisions, have been copied below.

Line 35, 'they persist up to now': "give date"

Line 54, 'after 2007': "I think this gives impression that springs only started in 2007"

Line 152, 'at present': "again give date/time – paper will go out of date!"

Line 157, 'diffusion': "isn't it really a hydrological diffusion – the actual fluid is flowing, not diffusing"

Line 162, 'up to now': "again give period"

Line 249, 'the differential equation': "need to define terms – this section needs better explanation – are just calculating adiabatic cooling or including conductive cooling?"

Reply to comments by Reviewer #1:

The ms is very interesting because it for the first time reports an anomalous metamorphic CO₂ degassing in association with a large earthquake in the Himalaya region. However in my opinion there are some (main) problems that should be fixed before an eventual publication in NC.

We are thankful to the reviewer for his/her positive comment concerning our work. We may also expand this comment because, to our knowledge, it is the first time that CO₂ degassing is affected by an earthquake in a non-magmatic region. We hope that our replies below and our associated modifications in the revised version will provide appropriate changes to address the problems pointed out by the reviewer.

1) I have some concerns about the use of the term 'eruption' (sometime 'massive CO₂ eruptions'!) referring to the phenomenology described in the ms. The described processes are very interesting especially for the geodynamic contest where they occur, but quantitatively are of minor relevance in comparison with other CO₂ emissions of the Earth. For example I note that the total CO₂ emitted by all the gas emissions in the region (estimated to be > 1x10⁸ mol/y, i.e. a few thousands tons of CO₂ per year) is generally much lower than the flux observed in other not volcanic areas of the Earth (even from single vents) (see, e.g., the data reported in Burton et al., 2013 or in Moerner and Etiope, 2002). For example in Italy a single emission of cold CO₂ from a not volcanic area discharges about 2000 tons of CO₂ per day (Chiodini et al., 2010), i.e. about 160 times higher than the estimated CO₂ flux from all the gas emissions in Nepal. In conclusion I suggest the authors to use a different terminology (i.e. emission instead of eruption) and to don't use the term "massive" or similar

In the submitted version of the manuscript, actually, we were attempting a parallel between our observations done in a non-volcanic region and some terminology used in volcanic environments. We thus opted in the title for the terms “eruption” and “unrest” to account for spectacular new or increased CO₂ emissions and for hydrothermal perturbations or changes in hydrothermal activity, respectively. We agree with the reviewer that the term “eruption” is a bit overstated, although it may be relevant when accounting for the large CO₂ outburst we observed in Sanjen. In the revised version, therefore, we changed “eruptions” into “emissions” in the title and throughout the manuscript, and into “outburst” at some selected occurrences, as suggested.

Concerning the adjective “massive”, we acknowledge that the total direct gaseous CO₂ discharges in Central Nepal that we are reporting here are two orders of magnitude smaller than other CO₂ degassing manifestations worldwide, in volcanic or magmatic context. However, the vocabulary used here should also be considered with respect to the context. Indeed, these new or increased CO₂ emissions are large compared with the other known CO₂ emissions in the Himalayas, and in this regard, the adjective “massive” may be relevant. To remain consistent, and not to add confusion, we removed it as suggested by the reviewer. We used the adjective “large” at some instances to highlight that these new CO₂ emissions are among the largest CO₂ emissions that were known before the earthquake. Finally, it should again be pointed out that, while the integrated discharges might not be considerable compared with an extended volcanic system, the local fluxes are extremely large (>10⁴ g m⁻² day⁻¹), in Syabru-Bensi (Upper Trisuli valley) or in the newly appeared zone in Machhakhola (Budhi Gandaki Valley), and larger than on numerous volcanoes.

Given the small intersect of active faults at the surface compared with volcanic vents, the observed values, and in particular the anomalous emissions that we had called eruptions, are extremely spectacular.

Concerning the example given by the reviewer, Mefite d'Ansanto in Italy, from which the etymology of the term "mofette" comes from, the system emits an extremely large amount of CO₂ in the atmosphere, but the gas bearing this CO₂ can also be considered of mantellic origin ($R/R_a \approx 2.7$). Here, in our case, we are dealing with non-volcanic and non-magmatic degassing as well as with a crustal source of the gas bearing CO₂ ($R/R_a < 0.05$). Nevertheless, we agree with the reviewer that, based on the current state of knowledge, the currently identified and studied Himalayan mofettes produce less CO₂ than the majority of volcanic and magmatic sites worldwide, but it should not be noted that they nevertheless remain in the same order of magnitude than other mofette sites in non-volcanic context (in the Eger Rift for example; Kämpf et al., 2013), as already shown for Syabru-Bensi by some of the authors (Girault et al., 2014a). Although the currently known direct gaseous CO₂ emissions of central Nepal cannot compete with active volcanic sites, many other CO₂ emitting spots are probably unknown yet all along the Himalayan arc.

When considering dissolved CO₂ in rivers and hot springs in addition to this direct CO₂ degassing, which is generally considered only as a small fraction of all the CO₂ emissions by the orogen, the extrapolated budget of CO₂ emission for the whole orogen is relatively large ($> 1.3 \times 10^{10}$ mol/year considering only central Nepal (Evans et al., 2008), and $> 10^{12}$ mol/year for the whole orogen (Becker et al., 2008)). In addition, from a petrological and petrographical point of view (Rapa et al., 2017), extrapolations of the metamorphic CO₂ production at the scale of the orogen lead to values in the range $0.3\text{--}4.4 \times 10^{11}$ mol/year. These extrapolated estimates may represent more than 10% of the total CO₂ emission by the solid Earth, which is not negligible and thus, definitely, worth of investigation. Finally, it should be pointed out that the zones of discharge observed till now might only be a small fraction of the total gaseous discharge. The discovery of gaseous discharges, when they are not clearly associated with a hot spring, as it is increasingly the case, is difficult and relies on a factor of luck. Without the hydropower project in the Sanjen valley, a large CO₂ emission triggered by the earthquake would have been totally missed. Thus, given the fact that we may have had access only to the tip of the iceberg till now, it might be adequate to remain cautious. For the moment, the most reliable global estimate remains the indirect inference from the water chemistry of hot springs and rivers, able to average over large drainage areas.

2) The authors does not give reliable proves that the increase in the degassing observed after the 2015 earthquake can not be due, at least partially, to different meteorological conditions that can heavily affect the soil diffuse degassing. The authors state "... These emissions showed remarkable temporal stability, even during monsoon17,..." The cited reference refers to a work that regards just one site and that, at least from the title, seems to deal with Radon instead of CO₂ (17: Richon, P. et al. Temporal signatures of advective versus diffusive radon transport at a geothermal zone in Central Nepal. J. Environ. Radioact. 102, 88-102 (2011)). It is worth to note that important seasonal (and daily) variations of intensity similar to the variation described in the work (e.g. factor 1.6) are normally observed in the sites where soil CO₂ fluxes are continuously measured (see for example Fig. 3 in Granieri et al. 2003 and Fig. 2 in Viveiros et al. 2014). Similar variations, that were observed in volcanic environments, could affect also the emission in Nepal because the variations do not depend on the origin of the gas but on meteorological conditions (i.e. atmospheric pressure changes, soil humidity, soil temperature etc.). I think that the authors should discuss how, also in the absence of continuous measurements, they can infer the deep origin of the observed degassing increases (for example because the contemporary occurrence in different location of the increase of CO₂

degassing after the earthquake, or because in some sites the measurements were done in the same period of the year, etc.)

We thank the reviewer for pointing out these matters. Here, two aspects are raised: the effect of meteorological perturbations and their possible consequences on our results, and the deep origin of the CO₂, partly related. We will reply in turn to these two questions.

First, concerning the meteorological perturbations, it should be pointed out that, in any case, the new measurements after the earthquake were performed during the best season for this type of measurement, namely the dry season. However, it turns out that we also had some detailed knowledge of the effect of monsoon before the earthquake, and not only concerning radon but also CO₂. These two points did need some clarification that we can support with some detailed work and additional information.

In the revised version, following the reviewer's comment, we now present in Fig. 3 the available 2015–2017 rainfall data in Dhunche and Timure, located seven kilometres southwest and nine kilometres north to Syabru-Bensi, respectively. These data are a courtesy of the Nepal Department of Meteorology, Kathmandu. The reader is now able to identify the monsoon periods and to check that no measurement was carried out during these periods. In the main text and in the Methods section, we now state clearly that no measurement was done during the monsoon periods or after a rainy episode. In Table 1 and Extended Table 1 for example, the month/year of all measurement campaigns was inserted. Moreover, the stability of these hydrothermal systems, even during the monsoon periods, was also attested by our published observations. Indeed, the seasonal and yearly stability of the soil-gas radon concentration time-series (Richon et al., 2011; Girault et al., 2014a), the invariant radon–CO₂ fluxes relationship (Girault et al., 2009; Girault et al., 2014b), and the results of watering experiments on CO₂ and radon fluxes (Girault et al., 2009) at selected sites accounted for the remarkable temporal stability of these hydrothermal systems. These last experiments showed that, even after a strong rainfall, the large CO₂ fluxes return to their pre-rainfall values a few hours only after the rain. Thus, the effects of real or artificial rain on CO₂ flux have been extensively studied before, with details in the published Supplementary Materials. The stability of the CO₂ fluxes in Syabru-Bensi and Timure, and the lack of a significant effect due to rainfall were actually a surprise then in 2008–2010, probably due to the fact that, locally, the fluxes are extremely high. During heavy monsoon rainfall, in Syabru-Bensi, heavy bubbling through rainwater ponds quickly takes place.

In any case, in the present paper, at all our studied sites, the CO₂ flux data were obtained before and after the earthquake at about the same periods under similar meteorological conditions. Besides, the observed changes in the CO₂ emission of a factor between 2 and 25 (Table 1) in Syabru-Bensi, Sanjen, Chilime, and Timure in the Upper Trisuli valley, and in Machhakhola in the Budhi Gandaki valley, were done contemporary. All these sites are separated by more than ten kilometres. All these care and observations preclude to these changes any environmental, meteorological or shallow origin, but suggest rather that the Gorkha earthquake modified some physical properties such as permeability of an over-pressurised CO₂ reservoir at crustal depth.

Concerning the variations of the fluxes with atmospheric pressure, it leads to variations faster than one hour. These effects are also discussed in the previous papers listed above. However, the source of high-frequency flux variations, at our sites and on volcanoes also, is not completely understood. The CO₂ flux seems to be released in intermittent bursts, rather than a continuous outflow. This phenomenon is hard to measure with accumulation chambers, so far poorly constrained, and therefore is not understood at this time while it might be an essential property at all sites where gaseous CO₂ is discharged. With the accumulation chamber, only some time-average flux can be measured, which still exhibits non-Gaussian fluctuations. In any case, given the large numbers of measurements in each site, such high frequency effects, while they may affect some isolated values, have little impact on the average values and the given estimates of discharges.

Concerning the depth of the CO₂ source, we comment again on it below when discussing the conceptual models. However, the reviewer is correct here again: one argument for a deep CO₂

source is the fact that we observe similar effect with similar isotopic anomalies over distances of the order of 10 km, indicating that the source must be at a depth of the same order of magnitude of the horizontal homogeneity scale. Another argument in favour of a deep source is the value of the isotopic anomaly, and the associated interpretation as metamorphic decarbonation, which cannot take place in the near subsurface. The argument for a deep source, however, is not completely linked to the temporal stability. Large areas of diffuse emissions could have lead to small average fluxes, sensitive to transient states of the subsurface due to water infiltration and so on. It is not what happens; the emission is concentrated in some places, which also, in addition, provides temporal stability, except during earthquakes, which is the whole point of our observations after the Gorkha earthquake.

In the revised manuscript, we have tried to clarify these matters when they arise, giving additional references and material when necessary.

3) The third problem regards the absence of a conceptual model explaining the occurrence of CO₂ emissions in association with thermal springs (see Fig. 1). The physical models proposed by the authors treat the two processes (CO₂ emission and thermal water discharges) separately but I think that a more realistic scenario should consider that the gas is separated by the thermal waters during their rise (and depressurization) from depth to the surface. What is the gas content of the thermal waters? What is the chemical and isotopic compositions of the gas and of the thermal waters? What changes were observed in the chemical and isotopic compositions of gas and waters?

The reviewer is thanked for his/her comment on the need of a conceptual model. We had alluded to some model in some sentences, but we agree that we need to be more transparent on the matter and clearly state our case and arguments. Consequently, in the revised version, the discussion starts with conceptual models and their arguments. Additional information from water chemistry is also now given.

The concept that the gas is separated from the thermal water is one concept only. This model is interesting, and definitely should not be ruled out, but, as such, it does not incorporate additional information, already discussed in previous papers. Indeed, gas discharge has been observed in Syabru-Bensi some distance away (500 m) from the main hot springs. In Timure also, the main gas discharge is not at the location of the hot spring. So, we had proposed previously the alternative concept that gas is incorporated to all crustal fluids from below, the hydrothermal circulations being just one way, much more efficient than others in numerous places, to transport the CO₂ to the surface. With the spectacular and unambiguous observations after the earthquake, gas reservoirs are revealed kilometres away from hot springs, substantiating the possibility that gas and hydrothermal circulations are independent, while sometimes together to make use of connected pathways. Note also that water is more efficient than gas to create permeability by hydro-fracturing. Thus, by consequence, gas appears at places associated with the hydrothermal waters. However, it is also in other places, as revealed by the Sanjen outburst. We don't consider this alternative model as the ultimate truth. However, we present it as a reasonable expansion of the current standard model to accommodate the observations accumulated since gaseous discharges were directly characterised and spectacularly enhanced by the new observations after the earthquake, observations that we claim are incompatible with a pure hydrothermal origin of the gas discharge. All these arguments are now clearly stated rather than being implicit.

Thus, following the reviewer's comment, in the revised version, we now present in the introduction section the accepted conceptual model of CO₂ degassing from thermal water bodies in the Himalayas. This conceptual model was inferred from chemical and isotopic data of rivers and hot springs (Becker et al., 2008; Evans et al., 2008) and was confirmed by the occurrence and carbon isotopic composition of the direct CO₂ emissions at selected site of Central Nepal (Girault et al., 2014b). In this pioneering model, CO₂ is produced at >5 kilometres depth by metamorphic

reactions including decarbonation at 10–25 kilometres depth. Then, CO₂ percolates through fracture networks, reaching the MCT shear zone, where it mixes with meteoric water, probably at few kilometres depth. At or near the water table, a fraction of CO₂ may degas, and water discharges eventually at the surface as thermal spring(s) with temperature between 60 and 80°C. Degassed CO₂ may also be transported faster to the surface through a dry network of interrelated faults without much interaction with hydrothermal circulations (Girault and Perrier, 2014). This conceptual model explains the occurrence of CO₂ emissions in association with hot springs.

In the revised version, we also addressed the request of the reviewer for more water chemistry and, consequently, we compiled all dissolved inorganic carbon (C_{DIC}) and isotopic composition ($\delta^{13}\text{C}_{\text{DIC}}$) data of hot springs of Central Nepal. These data are gathered in Table 2 and in Extended Data Table 2. Note that, in the submitted manuscript, the $\delta^{13}\text{C}$ and CO₂ concentration of the CO₂ emissions were already given in details in Extended Data Table 1 (total of 69 data). To give some deserved emphasis to these data, they are now also included in Table 1 in the main text. From January 2016 to January 2018, a total of 71 measurements of C_{DIC} and $\delta^{13}\text{C}_{\text{DIC}}$ were carried out after the earthquake. In Syabru-Bensi where the number of measurement is significant (n=41), C_{DIC} of the SBP0 hot spring is now plotted in Fig. 3. The C_{DIC} of the SBP0 hot spring increased by $29\pm 2\%$ after the mainshock and, in September 2017, returned to values measured before the mainshock, while Ca and Na concentrations remained similar within 5%. The $\delta^{13}\text{C}_{\text{DIC}}$ decreased also significantly. These observations suggest a larger amount of dissolved carbon in the SBP0 hot spring after the earthquake, which is compatible with the aforementioned increase in gaseous CO₂ emissions.

From the data, it appears that, even if CO₂ is pervading the crust and is only locally mixed with the hydrothermal circulations, as we suggest, the interaction with water, hydrothermal water or groundwater, is in any case an essential aspect to understand the isotopic anomalies. Such a detailed understanding is beyond the scope of the present paper, where we want to stress the effect of the earthquake. Nevertheless, in the revised manuscript, we have tried to detail a few points. For example, the variation after the earthquake of $\delta^{13}\text{C}_{\text{DIC}}$ versus the variation, or the lack of variation, of the $\delta^{13}\text{C}_{\text{gas}}$ needs to be pointed out. It might indeed suggest a constant input of a production source of CO₂ from below, while the dynamics of water is affected by the earthquake, maybe through a change of permeability. To clarify these matters, some research directions do emerge for the future.

Multiphase modelling of CO₂ degassing from a water body, with, among other, water discharge and carbon fractioning, in such a context characterised by high thermal gradient (we probably have 150°C between one and two kilometres depth; Derry et al, 2009), and considering also deformation processes such as, at least, crustal permeability changes, does not exist and is definitely beyond the scope of our paper and beyond the scope of the available data. By opposition to this global approach, in our manuscript, we relied on first-order qualitative and sometimes quantitative physical models best able to reproduce some of our observations. These models, while simple, allowed us to give first-order explanations of the observed natural processes using a set of known parameters (those we measured) and a relatively small number of unknowns. We however agree that such an integrative physical model would give important clues for long-term understanding of the CO₂ degassing in orogens, in particular in the Himalayas where hydrothermal systems are in near-critical conditions. Our work here can thus be seen as a first step toward this integrative approach.

In any case, we believe that our observations are crucial elements that need to be taken into account when trying to build some understanding of the CO₂ chemistry and transport in the context of the active faults in Nepal, where hydrothermal circulations are one important element, but only one among other important elements.

4) The variation in the gas flux are reported in a table (extended data table 1) where the changes are not easily readable. I suggest to report the variation (before and after) also in the map of Fig. 1 where the

authors could use different symbols for CO₂ emissions, CO₂+thermal water and thermal waters and different colours to indicate the variation (e.g., red where the fluxes increased, blue where are stable, green where the fluxes decreased, gray where no comparative data are available)

We thank the reviewer for this comment, and, indeed, to make the complexity of the matter more readily apparent to the reader, in the revised version, we have modified the Fig. 1 accordingly. We now represent: (1) hot springs, CO₂ emissions, and hot springs + CO₂ emissions as three different symbols, and (2) the post-seismic effects with different colours, depending on the positive effect, negative effect, absence of any significant change, or when it is unknown. In addition, a column stating the post-seismic effect(s) for every CO₂ emission site and hot spring has been added to the Extended Data Tables 1 and 2 (and to the Tables 1 and 2), respectively. The Fig. 1 (together with Tables 1 and 2) now clearly summarise the observations reported in the paper. We thank the reviewer for his/her inspiring comment that has definitely improved the manuscript and the presentation of our results.

5) References. I suggest the authors to read the review article "The Role of Fluids in Tectonic and Earthquake Processes" (Miller 2013) and in particular the chapter 4.5 "Documented Fluid-Driven Earthquake Sequences" where are reported a series of cases of fluids anomalies during earthquakes and to add relevant papers to the reference list of the ms.

We thank the reviewer for having pointed out this reference. Indeed, we used this relevant review article to complement our introductory section about the growing number of evidence for deep fluids controlling aftershock distribution (fluid-driven earthquakes). We now state that the observation of fluid-driven earthquakes (Miller, 2013) (mainly aftershocks) has been done in several tectonic contexts, including rifting (Noir et al., 1997; Miller et al., 2004; Terakawa et al., 2010), subduction (Waldhauser et al., 2010; Wang et al., 2012) reverse (Sibson, 2007) and strike-slip faulting (Cappa et al., 2009). No observation has been reported in the context of continental collision. Analogously, in the introduction section of the revised manuscript, we added some relevant papers reporting earthquake-induced effects on near-surface aquifers.

In conclusion I think that the ms is interesting but that in the present form can not be published. However, I think also that the authors could fix the above problems.

We thank the reviewer for his/her constructive comments that greatly improved the quality of our manuscript, and helped to better emphasize the major features of our results. Finally, we selected some of the references listed below by the reviewer, and cited them in the revised version.

Cited references

Burton, M. R., Sawyer, G. M. & Granieri, D. Deep carbon emissions from volcanoes. *Rev. Mineral. Geochem.* 75, 323–354 (2013).
Chiodini G, Granieri D, Avino R, Caliro S, Costa A, Minopoli C, Vilardo G (2010) Non-volcanic CO₂ Earth degassing: Case of Mefite d'Ansanto (southern Apennines), Italy. *Geophys Res Lett* 37:L11303, doi: 10.1029/2010GL042858
Granieri, D., G. Chiodini, W. Marzocchi, and R. Avino (2003), Continuous monitoring of CO₂ soil diffuse degassing at Phlegraean Fields (Italy): Influence of environmental and volcanic parameters, *Earth Planet. Sci. Lett.*, 212, 167–179.
Miller (2013), *The Role of Fluids in Tectonic and Earthquake Processes*. *Advances in Geophysics* 54:1–46 • December 2013
Mörner NA, Etiope G (2002) Carbon degassing from the lithosphere. *Global Planet Change* 33:185–203
Viveiros, F., J. Vandemeulebrouck, A. P. Rinaldi, T. Ferreira, C. Silva, and J. V. Cruz (2014), Periodic behavior of soil CO₂ emissions in diffuse

degassing areas of the Azores archipelago: Application to seismovolcanic monitoring, J. Geophys. Res. Solid Earth, 119, 7578–7597, doi:10.1002/2014JB011118.

Reference of the reply (not included in the manuscript):

Kämpf, H., Bräuer, K., Schumann, J., Hahne, K., Strauch, G., 2013. CO₂ discharge in an active, non-volcanic continental rift area (Czech Republic): Characterisation ($\delta^{13}\text{C}$, $^3\text{He}/^4\text{He}$) and quantification of diffuse and vent CO₂ emissions. Chem. Geol. 339, 71-83.

Reply to comments by Reviewer #2:

This is an interesting paper that presents a simple but important set of observations. There are a number of minor corrections needed to the manuscript which I have marked on the attached version.

We are thankful to the reviewer for his/her kind appreciation of our work. Although our observations may appear simple, probably because the measurements of temperature, fluxes and concentrations of gas and water can be relatively easily done in the field, we would like, in this manuscript, to emphasize the fact, pointed out by the reviewer, that this set of observations is unique in the Himalayas as well as in all other large orogens of the Earth, and thus provides an essential step forward to be able, in the future, to detect efficiently potential changes in the state of hydrothermal sites subject to crustal deformation. Our observations, however, are not as simple as they appear. Numerous sites are difficult to reach and proper portable equipment and methodologies, different from what is routinely available in volcanic observatories, had to be developed and tested. The quality of our data set results from a decade of hard work in difficult conditions and our ability to work repeatedly in numerous sites, to be able to discover new sites in remote areas, and to be constantly supported by the population through the hardships of a large deadly earthquake and, before that, a bloody insurgency. In the revised version, we have further emphasized the overall quality and uniqueness of our data-set and the importance of its use in such large orogens. We are also bringing new evidence and updates thanks to a detailed campaign carried out in January 2018, which brought slight modified view on the overall condition of the system. In some part, the current evidences suggest a return to the pre-earthquake equilibrium in some locations and in some parameters, while instability remains a persistent feature in other places, or in other parameters.

The explanation for the change in spring temperature in the methods section needs to be better explained.

Following the request of the reviewer, we have further expanded the description of our analytical model to relate temperature and flow rate of a hot spring. Additional explanations are also given in the Extended Data.

Line 35: "now" Give date.

According to our data-set and the new measurements carried out during our last field campaigns in September 2017 and January 2018, these new CO₂ emissions persist in January 2018, thus more than 2.7 years after the mainshock. In the Upper Trisuli valley, some of them (for example in Syabru-Bensi) have reached their maximum amplitude at the end of 2016 (Fig. 3), hence about 1.5 years after the mainshock, while others are still evolving actively showing significant increase (for example in Timure). The date is now systematically given in the main text and dates are stated in Table 1 and Extended Data Table 1.

Line 40: "episodic" instead of "unsteady".

We acknowledge that the adjective "unsteady" did not reflect our meaning. However, we do not agree with the reviewer, because "episodic CO₂ production" would mean that CO₂ production can also be inexistent, which is clearly not the case. Instead, we have preferred to use "non-stationary" in order to stress the fact that CO₂ production is not null, but can reach sometimes extreme (higher or smaller) values, as shown in our manuscript, after a significant earthquake.

Non-stationarity of the discharge can be due to non-stationarity of the production source, or to transient changes in the transport (permeability). Using "non-stationary" also characterizes better

the inherent out-of-equilibrium thermodynamic state of the faults-gas-hydrothermal system during the earthquake cycle, itself an intrinsically non-stationary perturbation and non-stationary symptom of mountain building processes.

Line 54: "After 2007" I think this gives impression that springs only started in 2007.

We agree with the reviewer that this phrasing may have lead to confusion. We decided to remove it in the revised version and, in general, we have tried to be more precise with all timing information, a crucial aspect of our observation and of the evolution of the system. First measurements are given in Table 1 and Extended Data Table 1, where the month and year of measurements are now given for each site.

Line 108: "now" Give date.

As commented above, in the revised version, to be more precise with timing information, we have removed "now" and, in this particular instance, added that in January 2018, hence >2.7 years after the earthquake, the flow of some of these new hot springs persists.

Line 152: "at present" Again give date/time - paper will go out of date!

Here also, as suggested by the reviewer, we have added the date of our last observations (January 2018). In general, date/time replaces the mention of "now" or "present".

Line 157: Isn't it really a hydrological diffusion - the actual fluid is flowing, not diffusing.

It is hydrogeological diffusion, which describes the transport of a pressure perturbation in a connected network or porous medium. Fluids always flow, but the perturbation obeys to a diffusion equation (actually a Boussinesq equation). Diffusivity considered here refers to one term of the equation, the dominating diffusive part, of the poroelastic effect. When dealing with phenomenological analysis of observations, diffusion-like processes, characterized for example by a penetration depth proportional to the square root of the characteristic time, involve a single parameter, whose value may not be necessarily easily interpreted, either of pure hydrogeological transfer or poroelasticity. Here, a detailed theoretical interpretation is beyond the scope of the paper, but we have tried to describe more clearly the actual processes that are discussed. We also provide an additional reference, just appeared, dealing with the modelling of fluid diffusion accounting for the delayed large aftershock, on May 12, of the April 25 Gorkha earthquake (Tung et al., in press).

Line 162: "up to now" Again give period.

As commented above, we have given the time period. In this instance, we have inserted that this linear trend in the GPS time-series is continuing up to our last available data in October 2017, hence about 2.5 years after the earthquake.

Lines 209 and 212: "Punctual" ??

At these two occurrences, the term "punctual uncertainty" refers to the experimental uncertainty that only depends on the measurement done in the field. The use of "punctual" is actually extremely precise, referring to statistically determined uncertainty on a given observation, to be opposed to systematic effects (such as calibration uncertainties or modulation by an external common source), which are common to all points of a given data set. It means that different and independent values of a given measurement should be distributed within a standard deviation compatible with the given punctual uncertainty. To avoid discussing such unnecessary details, in the revised manuscript, we are simply mentioning "experimental uncertainty".

Line 249: Need to define terms - this section needs better explanation - are just calculating adiabatic cooling or including conductive cooling?

Indeed we need to be more precise with the physical model. In the manuscript, therefore, some precisions are added in the main text and in the "Methods" section, and the model is described in extenso in the Supplementary Materials for the readers who cannot have access to the reference in French that we have been using. It is actually a simple model for the cooling of the hydrothermal circulations. Concerning the gas, we did not need to discuss the adiabatic cooling in our qualitative discussion, as the temperature distributions are largely dominated by conductive dissipation of the hydrothermal circulations and the locally enhanced geothermal gradient.

Minor comments:

Line 35: "the" intensity.
Line 37: "appearance" instead of "apparition".
Line 69: "from" instead of "in".
Line 69: "A quantitative comparison".
Line 70: "may be made" instead of "is meaningful".
Line 70: "because" instead of "thanks to".
Line 71: Insert "(Methods) were made before".
Line 78: "A smell".
Line 104: "shallow sources" instead of "that they are superficial effects".
Line 109: "The temperature".
Line 117: "spectacular change in hydrothermal activity" instead of "spectacular hydrothermal unrest".
Line 124: "stimulation".
Line 148: "a pre-seismic"
Line 151: "the shut down of the Chilime hot spring" instead of "Chilime hot spring cessation".
Line 152: "the permeability".
Line 153: "a vertical permeability".
Line 160: "the Sanjen eruption".
Line 163: "to changes in the state of" instead of "a slight change in state of".
Line 165: "are therefore".
Line 168: "currently" instead of "yet".
Line 170: "as a possible indication of" instead of "opening the door to possible observation of".
Line 185: "the monsoon".
Line 186: "CO₂ fluxes".
Line 190: "The CO₂ flux".
Line 210: "The pH of the water in thermal springs" instead of "Water pH of thermal springs".
Line 213: "The flow rate".
Line 217: "anomalies".
Line 223: "which" instead of "having".
Line 233: "A pressure source".
Lines 233-234: "a pressure distribution".
Line 234: "follows" instead of "obeys".
Line 234: "An example calculation".
Line 237: "described" instead of "recalled".

We are thankful to the reviewer for his/her thorough reading of our manuscript and his/her associated comments available in the pdf version. All these minor comments have been implemented in the revised version, as suggested.

REVIEWERS' COMMENTS:

Reviewer #1 (Remarks to the Author):

The authors answered positively to the suggestions and the manuscript is improved. In my opinion the manuscript is now ready to be published.

Giovanni Chiodini

Reply to comments by Reviewer #1:

The authors answered positively to the suggestions and the manuscript is improved. In my opinion the manuscript is now ready to be published.

Giovanni Chiodini

We are thankful to Giovanni Chiodini (Reviewer #1) for his positive appreciation of our work done to revise the manuscript. His thorough and inspiring comments have been extremely useful to improve the overall quality of our manuscript.